# A randomized stepped wedge trial of an intensive combination approach to roll back the HIV epidemic in Nigerian adolescents: iCARE Nigeria treatment support protocol

Babafemi O. Taiwo[1]*, Lisa M. Kuhns[2,3], Olayinka Omigbodun[4,5], Olutosin Awolude[6,7], Kehinde M. Kuti[7,8], Adedotun Adetunji[9], Baiba Berzins[1], Patrick Janulis[10], Sulaimon Akanmu[11], Oche Agbaji[12], Agatha N. David[13], Akinsegun Akinbami[14], Abiodun Folashade Adekambi[15], Amy K. Johnson[2,3], Ogochukwu Okonkwor[1], Bibilola D. Oladeji[9], Marbella Cervantes[3], Olubusuyi M. Adewumi[16], Bill Kapogiannis[17], Robert Garofalo[2,3]

1 Division of Infectious Diseases and Institute for Global Health, Northwestern University, Chicago, IL, Unites States of America, 2 Department of Pediatrics, Northwestern University Feinberg School of Medicine, Chicago, IL, Unites States of America, 3 Division of Adolescent Medicine, Ann and Robert H Lurie Children's Hospital of Chicago, Chicago, IL, Unites States of America, 4 Department of Child and Adolescent Psychiatry, and Centre for Child and Adolescent Mental Health, College of Medicine, University of Ibadan, Ibadan, Nigeria, 5 Department of Psychiatry, College of Medicine, University of Ibadan, Ibadan, Nigeria, 6 Department of Obstetrics and Gynecology, College of Medicine, University of Ibadan, Ibadan, Nigeria, 7 Infectious Disease Institute, College of Medicine, University of Ibadan, Ibadan, Nigeria, 8 Staff Medical Services Department, University College Hospital, Ibadan, Nigeria, 9 Department of Family Medicine, University College Hospital, Ibadan, Nigeria, 10 Department of Medical Social Sciences, Northwestern University, Chicago, IL, Unites States of America, 11 Lagos University Teaching Hospital, Lagos, Nigeria, 12 Department of Medicine, University of Jos and Jos University Teaching Hospital, Jos, Nigeria, 13 Nigerian Institute of Medical Research, Lagos, Nigeria, 14 Lagos State University College of Medicine, Ikeja, Nigeria, 15 Department of Paediatrics, Olabisi Onabanjo University and Olabisi Onabanjo University Teaching Hospital, Sagamu, Ogun State, Nigeria, 16 Department of Virology, College of Medicine, University of Ibadan, Ibadan, Nigeria, 17 Eunice Kennedy Shriver National Institute of Child Health and Human Development, Bethesda, Maryland, Unites States of America

* b-taiwo@northwestern.edu

**Data Availability Statement:** No datasets were generated or analyzed during the current study. All

## Abstract

### Background

Nigeria is one of six countries with half the global burden of youth living with HIV. Interventions to date have been inadequate as AIDS-related deaths in Nigeria's youth have remained unchanged in recent years. The iCARE Nigeria HIV treatment support intervention, a combination of peer navigation and SMS text message medication reminders to promote viral suppression, demonstrated initial efficacy and feasibility in a pilot trial among youth living with HIV in Nigeria. This paper describes the study protocol for the large-scale trial of the intervention.

### Methods

The iCARE Nigeria-Treatment study is a randomized stepped wedge trial of a combination (peer navigation and text message reminder) intervention, delivered to youth over a period of 48 weeks to promote viral suppression. Youth receiving HIV treatment at six clinical sites

relevant data from this study will be made available upon study completion.

**Funding:** This publication was supported by funding from the Eunice Kennedy Shriver National Institute of Child Health & Human Development (https://www.nichd.nih.gov) of the National Institutes of Health under Award Number UH3HD096920. The content is solely the responsibility of the authors and does not necessarily represent the official views of the National Institutes of Health under award number UH3HD096920 to BT and RG. The funding source has no role in the original design of this study, analysis and interpretation of data, or decision to submit results.

**Competing interests:** The authors have declared that no competing interests exist.

in the North Central and South Western regions of Nigeria were recruited for participation. Eligibility criteria included registration as a patient at participating clinics, aged 15–24 years, on antiretroviral therapy for at least three months, ability to understand and read English, Hausa, Pidgin English, or Yoruba, and intent to remain a patient at the study site during the study period. The six clinic sites were divided into three clusters and randomized to a sequence of control and intervention periods for comparison. The primary outcome is plasma HIV-1 viral load suppression, defined as viral load $\leq$ 200 copies/mL, in the intervention period versus the control period at 48 weeks of intervention.

## Discussion

Evidence-based interventions to promote viral load suppression among youth in Nigeria are needed. This study will determine efficacy of a combination intervention (peer navigation and text message reminder) and collect data on potential implementation barriers and facilitators to inform scale-up if efficacy is confirmed.

## Trial registration

ClinicalTrials.gov number, NCT 04950153, retrospectively registered July 6, 2021, https://clinicaltrials.gov/.

## Introduction

Since 2016, guidelines in Nigeria, Africa's most populous country, have recommended antiretroviral therapy (ART) regardless of CD4 cell count for all people with HIV [1]. Effective ART suppresses HIV-1 replication, which curbs AIDS-related morbidity and mortality, limits viral transmission, and is essential to end the epidemic [2–4]. Youth in Nigeria have not benefited from ART as much as adults [5–8]. The United Nations defines youth as individuals aged 15–24, in recognition of the developmental risks to health and well-being of mid-to-late adolescence and young adulthood [9] and has called for their prioritization in national HIV treatment strategies [10]. An estimated 180,000 youth ages 15–24 are living with HIV in the country, with 23,000 new infections in 2020 [11]. Viral load suppression is estimated to be approximately 43% across age groups, however, it is significantly lower among youth and younger adults (aged less than 34 years) [12]. In 2020, an estimated 3,000 youth died of AIDS-related causes in Nigeria [11]. Effective interventions to promote ART adherence and suppression of HIV-1 viremia are needed to ultimately end the epidemic among vulnerable youth.

Peer support, which shifts supportive care to community-based and non-professional providers for efficient use of resources [13, 14], is recommended by the World Health Organization as a strategy that provides many benefits for youth across the HIV care continuum [15]. Peer support can also motivate and reinforce self-care and help to build self-efficacy for HIV treatment [15]. In addition, mobile technology-based strategies (mHealth) have emerged as a promising approach to deliver HIV-related interventions, particularly among youth, with a growing body of evidence suggesting efficacy [16–18]. Approaches using mHealth have the advantage of simple interfaces for users, accessibility anywhere mobile telephone signals and/or Wi-Fi are available, relative affordability, and have been promoted specifically to reach stigmatized and disenfranchised populations [19, 20]. An SMS text message reminder intervention, the Treatment Text intervention (TXTXT), developed by members of our research team

has shown evidence of both feasibility and efficacy to promote ART adherence among youth with HIV in the United States (US) [21]. Based on social cognitive theory [22, 23], TXTXT is a bi-directional and personalized reminder intervention to promote medication adherence, which incorporates environmental cues to action and self-management support and feedback.

The iCARE Nigeria initiative includes both an HIV treatment support intervention and an HIV testing promotion intervention (parallel combination interventions). Herein, we describe the treatment support intervention of iCARE Nigeria (iCARE-Treatment), which incorporates locally adapted peer navigation and TXTXT to directly support youth living with HIV and promote viral suppression and is being tested via the UG3/UH3 mechanism, funded by the US National Institutes of Health (NIH). In the UG3 pilot phase of the study, which was completed prior to the initiation of the protocol described herein, iCARE-Treatment was piloted in a 48-week pre-post trial; participants (N = 40) were 50% male, mean age 19.9 years (range = 15–24), and 55% perinatally infected. Viral suppression (defined as viral load $\leq$ 200 copies/mL) was 35% at baseline and increased to 68% at 24 weeks and 60% at 48 weeks. These increases correspond to an odds ratio (OR) = 14.0 (p < .001) and OR = 6.0 (p = 0.013), at 24 and 48 weeks, respectively. Participants reported high satisfaction, with 100% indicating that they were satisfied/mostly satisfied with the intervention and 100% would refer a friend to receive it [24].

## Study objectives

The purpose of this study is to assess the efficacy of iCARE-Treatment, a combination peer navigation and text message medication reminder intervention, on viral suppression among youth living with HIV, aged 15–24. Secondary objectives are to assess intervention effects on ART adherence and retention in HIV care.

## Materials and methods

### Design

The protocol described herein is presented following the SPIRIT guidelines with related checklist (Fig 1). This study is a randomized stepped wedge trial among six clinical sites in Nigeria. A cluster-randomized design was chosen over individual-level randomization to eliminate the threat of contamination across intervention and control conditions at each site, given the social aspects of the peer navigation intervention. We chose a stepped wedge cluster randomized design because it allowed all participants to receive the intervention and has logistical advantages of staggering training and intervention launch between clusters over time, so that limited resources can be used strategically. The sites were divided into three clusters consisting of one, three, and two sites respectively. The Cluster 1 sites include the Infectious Disease Institute of the College of Medicine, University of Ibadan (IDI/CoMUI); Cluster 2 sites include Lagos State University Teaching Hospital (LASUTH), Lagos University Teaching Hospital (LUTH), Nigerian Institute of Medical Research (NIMR); and Cluster 3 sites include Jos University Teaching Hospital (JUTH) and Olabisi Onabanjo University Teaching Hospital (OOUTH). In addition, IDI/CoMUI and LASUTH have 1 and 3 affiliate clinics linked to the sites, respectively, and LUTH, NIMR, JUTH and OOUTH have 2 each. The study duration of 96 weeks consists of four 24-week periods, which include control (pre-intervention), intervention, or follow-up (post-intervention) conditions. Per the stepped wedge design (Fig 2), there are three different sequences: 1) two 24-week intervention periods then two follow-up periods (no control period); 2) one 24-week control period followed by two intervention periods then one follow-up period; 3) two 24-week control periods followed by two intervention periods (no follow-up period). Thus, each sequence includes 48 continuous weeks of the iCARE-treatment intervention, which consists of peer navigation and support combined with daily text

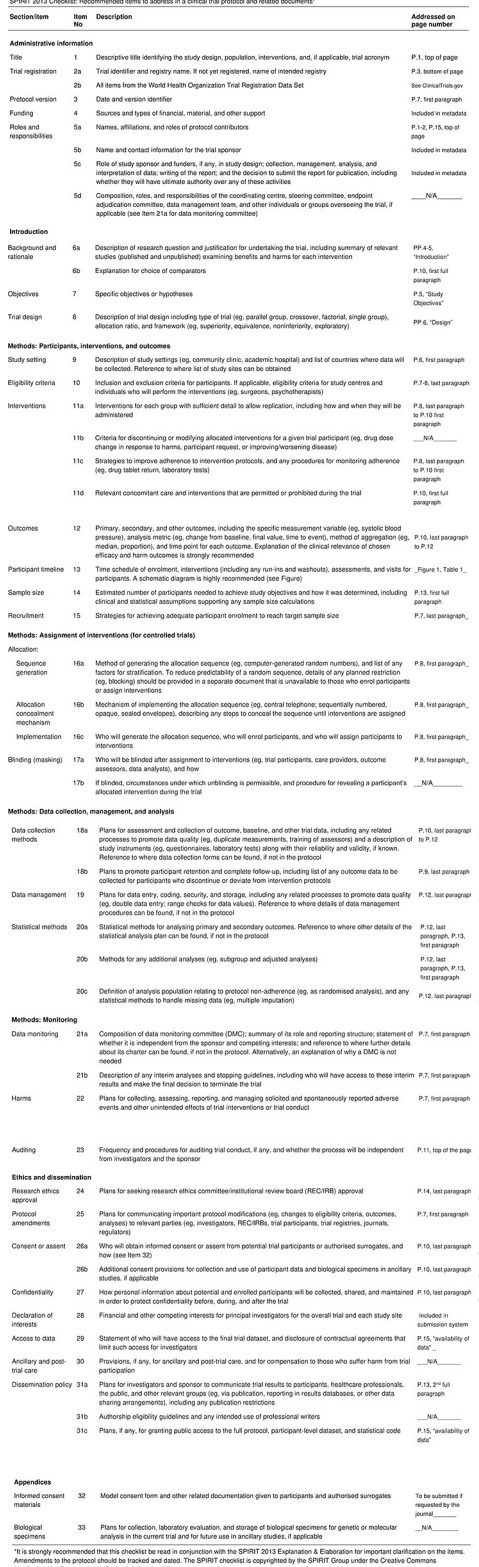

**Fig 1. SPIRIT checklist.**

| STUDY WEEKS | 0-24W | 25-48W | 49-72W | 73-96W |
|---|---|---|---|---|
| TIME PERIOD | TIME 1 | TIME 2 | TIME 3 | TIME 4 |
| Cluster 1: IDI/CoMUI (not randomized) | | | | |
| Cluster 2: RANDOMIZED (LASUTH, LUTH, NIMR) | | | | |
| Cluster 3: RANDOMIZED (JUTH, OOUTH) | | | | |

| | |
|---|---|
| Control (Pre-Intervention) Phase | |
| Intervention Phase | |
| Follow-up (Post-Intervention) Phase | |

**Fig 2. iCARE study schema.**

message medication reminders (TXTXT). The control and follow-up periods are observational, with collection of data only, while receiving standard care. Study enrollment and data collection began in April of 2021 and all data are expected to be completed by September of 2023.

## Ethical approval

This protocol has been approved by the Institutional Review Boards of Northwestern University (STU00214317-CR0001), Ann & Robert H. Lurie Children's Hospital of Chicago, College of Medicine, University of Ibadan/University College Hospital; Lagos State University Teaching Hospital, Lagos University Teaching Hospital, Nigerian Institute of Medical Research; Jos University Teaching Hospital and Olabisi Onabanjo University Teaching Hospital respectively, with a waiver of parental permission for participation of minors (aged 16–17). Current protocol version: March 11, 2022. Any modifications to the protocol are submitted to and approved by the IRBs of Record prior to implementation. Spontaneously reported adverse events and unintended effects of the trial are tracked by the study Principal Investigators and reported to the IRBs of Record. Study participants complete a written informed assent or consent process prior to participation in research activities. This study is monitored by a Data Safety and Monitoring Board (DSMB), the composition of which may be obtained from the

study Principal Investigator at the lead institution. The trial will be suspended or stopped if recommended by the DSMB and the study investigators, in consultation with the study sponsor, determine that, in their clinical opinion, there are risks to participants for continuation.

## Identification and recruitment of participants

Participants were recruited from youth patients at each clinical site. Individuals were eligible if they were: (a) living with HIV, (b) a registered patient at a clinical site, (c) aged 15 to 24 years (emancipated or with parental permission if aged 15), (d) on ART for at least three months and (e) able to understand and read basic English, Hausa, Pidgin English or Yoruba, (f) willing and able to provide informed assent or consent, and (g) intending to remain a patient in the study clinic during the study period (up to 96 weeks). Individuals were excluded if they were: (a) unable to provide informed assent or consent; (b) aged 15 years and are not emancipated or do not have parental permission for participation.

## Randomization

After the completion of initial enrollment, clusters were randomized to a sequence. We felt that cluster randomization via the stepped wedge design was reasonable in this case because the restricted age range and similar settings (university-based healthcare setting) were expected to make the clusters relatively homogenous. The random assignments were generated by a computerized random number generator by the Statistician (PJ). Randomized sequences were concealed from participants and study staff until enrollment was completed. Clusters 2 and 3 were randomized to the sequence with one or two initial control periods (data collection only), respectively, followed by two intervention periods. In order to run interim analyses by cluster and step, the statistician will not be blinded during statistical analyses.

Cluster 1 was not randomized but began the intervention at the point of enrollment, which was purposeful as this was a test site for the original UG3 pilot trial of the intervention and thus had staff ready and trained to begin the intervention at the start of the study period. The final sequence of the four 24-week periods in each cluster is shown in Fig 2. A two-day training was conducted with investigators and staff of clusters 2 and 3 after their initial control period (s), immediately prior to intervention deployment (staggered by start date). This was then followed by intensive step-down trainings at each cluster site with all staff and peer navigators (including alternate/replacement peer navigators), to prepare them to deliver the intervention.

## Description of the intervention period: iCARE-Treatment

The iCARE-Treatment intervention includes peer navigation and daily text message medication reminders. After assessment of the participant's needs and development of an action plan, peer navigators make at least one contact (in-person or phone) every two weeks with participants to support goals of the action plan. Peer support is designed to be dynamic and flexible, with support provided via text message, WhatsApp message, and phone calls and includes accompaniment to the clinic, assistance with medication pick-up, and other health-related appointments, as needed. Peer navigators also work with the study coordinator, project manager and investigators to facilitate referrals to mental health and other services and provide general peer support. Peer navigators are selected based on recommendations from their HIV care providers. Requirements for this role include living with HIV and being virologically suppressed in the prior 12-month period (or the last available viral load measurement), clinically stable according to their physician's subjective assessment, and aged 18–30 years. Age range and other targeted attributes of peer navigators are informed by findings from formative research conducted in youth and other stakeholders prior to the UG3 pilot study. Based on

findings from the UG3 adaptation and pilot study [25], we expected to match each peer navigator to 4–5 study participants with salient shared characteristics, such as gender, age group, residential location, education level, religion, and other social characteristics (e.g., marital status, motherhood). Peer navigators receive a monthly stipend to cover cost of transportation and telephone/text messaging plus honorarium for their time. The total amount provided as stipend is based on results of our preparatory stakeholder engagement activities and consistent with local standards.

The text message reminder intervention is an mHealth intervention using the TXTXT platform, designed and tested with evidence of efficacy in our prior work in the US [21, 26]. This intervention includes a youth-centered, personalized, and bidirectional set of three daily messages to support adherence. The daily text messages include a personalized initial medication reminder that coincides with the participant's chosen time to take their antiretroviral regimen, followed by a second message 15 minutes later, asking whether they took their medication, with a request for reply ("yes" or "no"). An encouraging third automated message, from our text message library, is then triggered by their reply. Both the initial and follow-up messages are chosen by each youth and personalized to reflect content meaningful to them. To protect confidentiality, messages are worded in a way that would not reveal the participant's HIV status if read by a third party. In the UG3 study, youth living with HIV worked with study staff to adapt the text message library to ensure it contained expressions understood and commonly used by youth locally. The text messages are delivered via the Dimagi Commcare platform (https://www.dimagi.com/commcare/). Participants use their own phones for text messaging; however, a phone is provided to participants who do not have access to a phone, which is anticipated to be less than 20% of participants. Participants also receive 100 Nigerian Naira per week during the intervention period to cover the (minimal) costs of text messages (i.e., "airtime").

## Control period

During the pre-intervention control periods, study participants receive standard-of-care (SOC) adherence counseling only. In Nigeria, per national guidelines [27], SOC adherence counseling includes a readiness assessment at initiation of ART to emphasize both the benefits of medication adherence and also the consequences of poor adherence, including virological failure; the development of a patient-centered adherence strategy, and on-going monitoring. Two consecutive viral load test results above 1000 copies/mL, at least 3 months apart, triggers enhanced adherence counseling which includes exploration of specific barriers to adherence and development of an adherence intervention plan with follow-up at 3-month intervals.

## Follow-up period

This is the post-intervention period when daily text messaging and protocol-specified peer navigation are discontinued, and participants return to standard care.

## Study assessments

Participants are consented by study staff at the enrollment visit and complete interviewer-administered computerized study assessments at each visit, at 24-week intervals, captured in REDCap (see Table 1 for full schedule of activities). Study consent includes permission for use of participant data and samples for future studies. The longitudinal module of REDCap with survey queue and scheduling modules is used for both survey data collection and entry of data from case report forms, anchored to the enrollment date. All study data are securely stored at Northwestern University in a limited access database by study ID. All hard copy participant

**Table 1.  iCARE Nigeria treatment intervention schedule of activities.**

| | Pre-intervention (Control Sites) | Baseline (Intervention Phase) Intervention | Week 24- Intervention Phase | Week 48 – Intervention Phase | Post-intervention |
|---|---|---|---|---|---|
| Informed consent | X | X | | | |
| Blood draw for viral load | X | X | X | X | X[a] |
| Blood draw for CD4 count | X[b] | X[b] | | | |
| Blood draw for future studies (stored; virologic resistance testing and other studies) | X | X | X | X | |
| Adherence Assessment: Dried Blood Spot (DBS), Pharmacy Pick-up Records Abstraction [28, 29], Self-Report [30] | X | X | X | X | X[c] |
| Computer Assisted Personal Interviewing Questionnaires [d] | X | X | X | X | |
| Implementation Assessment [e] | X | X | X | X | X |
| Cost Assessment [f] | X | X | X | X | X |
| **SMS + Peer Navigation** | | Baseline through week 48 | | | |
| SMS Text Initiation | | X | | | |
| Peer Navigation Enrollment | | X | | | |

[a] Per medical chart abstraction of standard of care viral load quantification

[b] CD4 quantification will be performed only at the first study visit (either pre-intervention in Jos, Lagos, Sagamu), or baseline (in Ibadan)

[c] Medical records abstraction only (no self-report, no DBS)

[d] Computer assisted personal interviewing (CAPI): Demographic Characteristics, HIV treatment knowledge [31], HIV medication self-efficacy [32], HIV stigma [33], depressive symptoms [34], alcohol and drug use [35], intervention satisfaction and acceptability [36]

[e] RE-AIM framework implementation strategies [37], contextual factors from the Consolidated Framework for Implementation Research (CFIR) [38].

[f] Includes abstraction of cost data from site administrative records and through interviews with stakeholders

information (e.g., study checklists, consent forms) are securely stored at each study site in locked file cabinets with limited access. Study enrollment and data collection procedures are monitored annually by contractual arrangement with a third-party monitor, arranged by the study sponsor.

**Primary outcome.**   Viral load suppression is defined as plasma viral load < 200 copies/ mL. Blood for viral load quantification is collected via venipuncture at each study visit. Viral load quantification is performed at study laboratories using real-time HIV-1 COBAS® assay (Roche Molecular Systems, Inc.) with a lower detection limit of 20 copies/mL.

**Secondary outcomes.**   Retention in HIV care is defined as at least two medical visits in a 24-week study period [39], with clinic visits abstracted from the medical record. ART adherence is measured via self-report, pharmacy records, and quantification of antiretroviral drug levels in dried blood spots (DBS). Self-reported adherence is measured as 30-day adherence on a visual analogue scale (VAS) of 0 to 100 [30]. Pharmacy records are captured as part of routine care in the electronic medical record. Drug pick-up adherence based on medication possession ratio (MPR) will be calculated from pharmacy records [28, 29], using drug pick-up records over each 24-week period, as total number of days' supply obtained including the last fill, divided by number of days between first fill and last day of the observation window [40]. Five 50µL blood spots are collected on protein saver cards (DBS) for measurement of antiretroviral drug levels. The samples are dried out at room temperature for 24 hours, then sealed tightly with desiccant and a humidity indicator, and stored at -20 degrees Celsius until analysis. Clinic visit and pharmacy records are abstracted from medical records onto case report forms and entered in REDCap. Both VAS and MPR values will be evaluated as dichotomous indicators of ≥90% adherence (good) or < 90% adherence (poor) [41].

## Statistical analysis

Several data quality checks will be conducted prior to analysis, including analysis of descriptive statistics and graphic plots to detect appropriate range of variables and to detect missing data or invalid cases. All data are stored in a password-protected network server with daily back-up. For descriptive statistics, percentages will be presented for dichotomous outcomes (e.g., viral suppression vs. non-suppression) and means will be presented for continuous outcomes (e.g., medication adherence). Similarly, changes between intervention and control periods will be summarized via odds ratios. Generalized estimating equations (GEE) will be used to account for multiple observations of participants. A binomial distribution and logit link function will be used to evaluate the statistical significance of the intervention effect (i.e., difference in viral suppression between intervention and pre-interventions periods), while controlling for secular trends in the outcomes over time and accounting for cluster with fixed effects. This analysis will follow the principles of intention-to-treat analysis and will be evaluated by examining the statistical significance of the regression coefficient for the intervention on the primary outcome. For differences in viral load suppression, we will compare intervention and control periods at the end of the intervention period, i.e., 48 weeks. Data collected during the follow-up period will be analyzed to explore the durability or "post-dose effect" of the intervention.

## Sample size calculation

With a small to moderate within cluster correlation (ICC = 0.10) and moderate individual autocorrelation (0.50), this analysis will have adequate power (power = 0.945) to detect an OR of 1.5 in the proportion of participants achieving viral suppression with at least 90 participants per site. Our effect size estimate is conservative given that the iCARE UG3 pilot trial findings (OR = 14.0) was not randomized and not controlled. Findings from our prior randomized trial of text messaging on medication adherence (a related outcome) among YLH in the United States suggested an effect size of OR = 2.12 (95% CI 1.01–4.45).

Primary study data will be analyzed as soon as possible after the end of data collection, with study findings disseminated in peer-review public health journals.

## Discussion

We describe herein the design of a stepped wedge trial of the iCARE-Treatment intervention, a combination of peer navigation and text message reminders to promote viral suppression among youth living with HIV in Nigeria, aged 15–24 years. The intervention draws on evidence-based intervention approaches that we adapted to the Nigerian context through broad stakeholder input in the formative stage of study design. Following the pre-post pilot trial in the UG3 phase of this study with evidence of initial efficacy and feasibility, we sought to extend the evidence from the UG3 to a much larger sample in the UH3 study described herein. The design of this UH3 study has several additional strengths, including inclusion of youth living with HIV in multiple states and regions of Nigeria, using a practical stepped wedge design that ensures every cluster receives the intervention, and incorporating a combination of evidence-based intervention approaches.

iCARE-Treatment is among the first interventions to be tested in a rigorous trial of sufficient size to detect effects on ART outcomes in youth. Effective interventions are needed urgently, particularly in a country like Nigeria that has met none of the key metrics recommended by Joint United Nations Programme on HIV/AIDS (UNAIDS) for measuring progress in reducing the public health threat of HIV [11, 42]. Illustratively, AIDS-related deaths from 2010 to 2020 fell by only 28% compared to the target of 75% decline [11]. Since youth are

known to fare worse than older persons on the cascade of HIV care [43, 44], interventions like iCARE-Treatment are needed to improve viral suppression in youth; sustained viral suppression is the gateway to lower morbidity and mortality and elimination of viral transmission [45–47]. In designing this UH3 trial, our team is mindful of the need to understand potential barriers to broad adoption and scale-up of the iCARE-Treatment intervention should the results from the UG3 pilot be replicated in the UH3 study. As such, we intend to collect data on potential implementation barriers and facilitators, including cost implications, during the UH3 study and provide public health collaborators and policy makers with a comprehensive summary of evidence at the end of the study.

## Supporting information

**S1 File.**
(DOCX)

## Acknowledgments

We would like to thank members of the iCARE team at all study sites for their contribution to the study protocol and procedures.

## Author Contributions

**Conceptualization:** Babafemi O. Taiwo, Lisa M. Kuhns, Robert Garofalo.

**Funding acquisition:** Babafemi O. Taiwo, Robert Garofalo.

**Supervision:** Olayinka Omigbodun, Olutosin Awolude, Kehinde M. Kuti, Adedotun Adetunji, Sulaimon Akanmu, Oche Agbaji, Agatha N. David, Akinsegun Akinbami, Abiodun Folashade Adekambi.

**Writing – original draft:** Babafemi O. Taiwo, Lisa M. Kuhns, Robert Garofalo.

**Writing – review & editing:** Babafemi O. Taiwo, Lisa M. Kuhns, Olayinka Omigbodun, Olutosin Awolude, Kehinde M. Kuti, Adedotun Adetunji, Baiba Berzins, Patrick Janulis, Sulaimon Akanmu, Oche Agbaji, Agatha N. David, Akinsegun Akinbami, Abiodun Folashade Adekambi, Amy K. Johnson, Ogochukwu Okonkwor, Bibilola D. Oladeji, Marbella Cervantes, Olubusuyi M. Adewumi, Bill Kapogiannis, Robert Garofalo.

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
