## [Decision Letter · Decision Letter 0]

13 Dec 2022

PONE-D-22-22897A randomized stepped wedge trial of an intensive combination approach to roll back the HIV epidemic in Nigerian Adolescents: iCARE Nigeria treatment support protocolPLOS ONE

Dear Dr. Kuhns,

Thank you for submitting your manuscript to PLOS ONE. After careful consideration, we feel that it has merit but does not fully meet PLOS ONE’s publication criteria as it currently stands. Therefore, we invite you to submit a revised version of the manuscript that addresses the points raised during the review process.

We look forward to receiving your revised manuscript.

Kind regards,

Joel Msafiri Francis, MD, MS, PhD

Academic Editor

PLOS ONE

Journal Requirements:

This publication was supported by funding from the Eunice Kennedy Shriver National Institute of Child Health & Human Development of the National Institutes of Health under Award Number UH3HD096920. 

However, funding information should not appear in the Acknowledgments section or other areas of your manuscript. We will only publish funding information present in the Funding Statement section of the online submission form. 

This publication was supported by funding from the Eunice Kennedy Shriver National Institute of Child Health & Human Development (https://www.nichd.nih.gov) of the National Institutes of Health under Award Number UH3HD096920. The content is solely the responsibility of the authors and does not necessarily represent the official views of the National Institutes of Health under award number UH3HD096920 to BT and RG. The funding source has no role in the original design of this study, analysis and interpretation of data, or decision to submit results.

5. We note that the original protocol file you uploaded contains a confidentiality notice indicating that the protocol may not be shared publicly or be published. Please note, however, that the PLOS Editorial Policy requires that the original protocol be published alongside your manuscript in the event of acceptance. Please note that should your paper be accepted, all content including the protocol will be published under the Creative Commons Attribution (CC BY) 4.0 license, which means that it will be freely available online, and any third party is permitted to access, download, copy, distribute, and use these materials in any way, even commercially, with proper attribution.

Therefore, we ask that you please seek permission from the study sponsor or body imposing the restriction on sharing this document to publish this protocol under CC BY 4.0 if your work is accepted. We kindly ask that you upload a formal statement signed by an institutional representative clarifying whether you will be able to comply with this policy. Additionally, please upload a clean copy of the protocol with the confidentiality notice (and any copyrighted institutional logos or signatures) removed.

Reviewers' comments:

Reviewer's Responses to Questions

**Comments to the Author**

1. Does the manuscript provide a valid rationale for the proposed study, with clearly identified and justified research questions?

Reviewer #1: Partly

Reviewer #2: Yes

2. Is the protocol technically sound and planned in a manner that will lead to a meaningful outcome and allow testing the stated hypotheses?

Reviewer #1: Partly

Reviewer #2: Yes

3. Is the methodology feasible and described in sufficient detail to allow the work to be replicable?

Reviewer #1: No

Reviewer #2: Yes

4. Have the authors described where all data underlying the findings will be made available when the study is complete?

Reviewer #1: Yes

Reviewer #2: Yes

5. Is the manuscript presented in an intelligible fashion and written in standard English?

Reviewer #1: Yes

Reviewer #2: Yes

6. Review Comments to the Author

You may also provide optional suggestions and comments to authors that they might find helpful in planning their study.

Reviewer #1: I have read with interest this submission whose intention seems to publish the protocol of a future cluster randomized trial based on a previous smaller pilot study investigating the same intervention.

Although this type of submission should be encouraged it also represents a stage at which the trial design might be questioned by external reviewers. I have the following requests of clarifications:

Major points

1. The timing is unclear. When is time 0 likely to be for the earliest cluster to enter the study and over which time-span is the RCT planned to occur?

2. Primary endpoint. A HIV-1 COBAS® assay (Roche Molecular Systems, Inc.) with a lower detection limit of 20 copies/mL is going to be used to measure viral load. Unclear why a threshold d of 200 copies is used to establish viral suppression instead. Also unclear from Table 1 whether blood will be drawn after control periods and follow-up periods. These viral loads are crucial for the comparison with the values measured after intervention.

3. Unclear why the statistical unit in the trial are the 3 clusters and not the clinical site. If I understood well ultimately Cluster 2 will be equivalent to sequence 2 and Cluster 3 to sequence 3. It would make more sense to me that each site (apart from site #1 for which I appreciate the rationale for being allocated deterministically to sequence 1) have equal chance to be allocated either to sequence 2 or sequence 3. In general, there the need to clarify the advantages of such a cluster trial randomisation as opposed to individual participants randomisation.

4. It is not uncommon that at least one statistician in the trial team is not blind to the randomisation. The justification given (the complexity of the analysis) is however unusual. I assume that the authors mean that one statistician will remain un-blind to be able to produce interim analysis reports by clusters and sequence. Please clarify.

5. More details are needed to describe the creation of randomisation schedules. What kind of randomisation will be employed? The number of clusters available is limited and therefore simple forms of randomization may not achieve balance between intervention and control arms at either the cluster- or participant-level.

6. Unclear why such a small OR of 1.5 was used to calculate statistical power. The pilot study indicates that the magnitude of the effect is likely to be a lot larger and up to 14-fold difference. Also unclear how the individual correlation (r=0.5) and cluster-correlation (r?? <0.5) estimates are coming from.

7. Statistical analysis section is quite short and has little details/clarity. Because of the use of GEE methods my understanding is that the strategy will be compared only within clusters (so that each participant act as his/her own control). Will be any comparison between sequences? If so, what is going to be the approach to analysis? Clusters can be considered independent and therefore GEE should not be needed.

8. The trial is going to be costly. I am not an expert in the field but is there a plan to evaluate whether the possible benefit in terms of increased number of youths with a suppressed viral load as a consequence of the intervention is going to be cost-effective? There is only a hint to this possible analysis in the last paragraph of the Discussion.

9. One of the possible advantages of publishing a trial protocol is aiming to pre-register the study with a journal prior to data collection/collation, to avoid publication bias and limit researcher degrees of freedom. Authors should clarify whether this is an intent as it would strengthen the submission.

Minor points

1. Wording is fairly confusing with some of the sentences phrased in the future ‘it is anticipated that…’ ‘participants will be….’ ‘quality checks will be conducted’ ‘data will be analysed’ etc. vs. others which are placed in the past (like the trial has actually already occurred) e.g. ‘Cluster 1 was not randomised’ ‘participants were recruited’ etc. Suggest phrasing is standardised throughout.

2. Lines 90-96. Form the way the paragraph is structured, it is not obvious at first reading that these are the results of a separate pilot study and not those produced by preliminary data of the current protocol.

3. Line 95. Please indicate exact p-value instead of <0.05.

Reviewer #2: The authors present a description of the methods for a cluster randomized stepped wedge trial of an intervention to increase viral suppression on HIV-1 antiretroviral treatment targeted towards youth. The protocol is clear and well written and is presented in a manner that others could replicate. I have no substantive edits for the authors. Thanks for following the SPIRIT guidelines.

Minor comment:

Based on the results of the pilot, it would appear the trial is overpowered if the target is 1.5 times the odds of viral suppression comparing pre to post implementation. The authors might want to consider what is a clinically important increase in viral load suppression (although anything up from the baseline in the 30's would be better) rather than the 1.5 which would not result viral suppression levels that are beneficial at a the target population level.

It appears the population for the intervention is all youth regardless of mode of acquisition of HIV infection. Do the authors anticipate any difference in response to the intervention by youth who were perinatally infected versus infected as adolescents or young adults?

7. PLOS authors have the option to publish the peer review history of their article (what does this mean?). If published, this will include your full peer review and any attached files.

Reviewer #1: No

Reviewer #2: No

---

## [Author Response · Author response to Decision Letter 0]

10 May 2023

Journal Requirements: 

RESPONSE: We have revised the headings and subheadings to conform to the required style.

This publication was supported by funding from the Eunice Kennedy Shriver National Institute of Child Health & Human Development of the National Institutes of Health under Award Number UH3HD096920. 

However, funding information should not appear in the Acknowledgments section or other areas of your manuscript. We will only publish funding information present in the Funding Statement section of the online submission form. 

This publication was supported by funding from the Eunice Kennedy Shriver National Institute of Child Health & Human Development (https://www.nichd.nih.gov) of the National Institutes of Health under Award Number UH3HD096920. The content is solely the responsibility of the authors and does not necessarily represent the official views of the National Institutes of Health under award number UH3HD096920 to BT and RG. The funding source has no role in the original design of this study, analysis and interpretation of data, or decision to submit results.

RESPONSE: We have removed the funding information from the acknowledgements statement and the funding statement that you have on file is correct. 

RESPONSE: We have moved the ethics statement to the last paragraph of the methods section.

RESPONSE: There are no SI files included in this submission.

5. We note that the original protocol file you uploaded contains a confidentiality notice indicating that the protocol may not be shared publicly or be published. Please note, however, that the PLOS Editorial Policy requires that the original protocol be published alongside your manuscript in the event of acceptance. Please note that should your paper be accepted, all content including the protocol will be published under the Creative Commons Attribution (CC BY) 4.0 license, which means that it will be freely available online, and any third party is permitted to access, download, copy, distribute, and use these materials in any way, even commercially, with proper attribution.

Therefore, we ask that you please seek permission from the study sponsor or body imposing the restriction on sharing this document to publish this protocol under CC BY 4.0 if your work is accepted. We kindly ask that you upload a formal statement signed by an institutional representative clarifying whether you will be able to comply with this policy. Additionally, please upload a clean copy of the protocol with the confidentiality notice (and any copyrighted institutional logos or signatures) removed.

RESPONSE: The study sponsor does not prohibit or restrict publication of the protocol. The confidentiality statement appears in the protocol to prevent publication without permission. In the cover letter for this resubmission, we have granted permission for the publication of the protocol. We have also uploaded a clean copy of the protocol with the confidentiality notice removed. 

RESPONSE: We have reviewed the reference list to ensure that it is complete and correct.

Reviewers' comments:

Reviewer's Responses to Questions 

Comments to the Author

1. Does the manuscript provide a valid rationale for the proposed study, with clearly identified and justified research questions?

Reviewer #1: Partly

Reviewer #2: Yes

2. Is the protocol technically sound and planned in a manner that will lead to a meaningful outcome and allow testing the stated hypotheses?

Reviewer #1: Partly

Reviewer #2: Yes

3. Is the methodology feasible and described in sufficient detail to allow the work to be replicable?

Reviewer #1: No

Reviewer #2: Yes

4. Have the authors described where all data underlying the findings will be made available when the study is complete?

Reviewer #1: Yes

Reviewer #2: Yes

5. Is the manuscript presented in an intelligible fashion and written in standard English?

Reviewer #1: Yes

Reviewer #2: Yes

6. Review Comments to the Author

You may also provide optional suggestions and comments to authors that they might find helpful in planning their study.

Reviewer #1: I have read with interest this submission whose intention seems to publish the protocol of a future cluster randomized trial based on a previous smaller pilot study investigating the same intervention.

Although this type of submission should be encouraged it also represents a stage at which the trial design might be questioned by external reviewers. I have the following requests of clarifications:

Major points

1. The timing is unclear. When is time 0 likely to be for the earliest cluster to enter the study and over which time-span is the RCT planned to occur?

RESPONSE: Study enrollment and data collection began in April of 2021 and all data collection is expected to be complete by September of 2023. We have added this statement to the design section on p.6.

2. Primary endpoint. A HIV-1 COBAS® assay (Roche Molecular Systems, Inc.) with a lower detection limit of 20 copies/mL is going to be used to measure viral load. Unclear why a threshold d of 200 copies is used to establish viral suppression instead. Also unclear from Table 1 whether blood will be drawn after control periods and follow-up periods. These viral loads are crucial for the comparison with the values measured after intervention.

RESPONSE: The threshold of 200 copies/mL was chosen because this is the cut-off used to define virologic failure clinically. Plasma viral load values between 20 and 200 are thought to represent release of non-replicating viral particles that is not indicative of antiretroviral treatment failure. Consistent with this, undetectable viral load is defined as viral load below 200 copies/mL in the undetectable = untransmissible tenet (i.e., U=U). Blood will be drawn at all follow-up visits in both the control and follow-up periods. 

3. Unclear why the statistical unit in the trial are the 3 clusters and not the clinical site. If I understood well ultimately Cluster 2 will be equivalent to sequence 2 and Cluster 3 to sequence 3. It would make more sense to me that each site (apart from site #1 for which I appreciate the rationale for being allocated deterministically to sequence 1) have equal chance to be allocated either to sequence 2 or sequence 3. In general, there the need to clarify the advantages of such a cluster trial randomisation as opposed to individual participants randomisation.

RESPONSE: To clarify, clusters 2 and 3 did have equal chance of allocation to sequences 2 and 3, respectively. In terms of advantages, the stepped wedge trial in this case had logistical advantages in that training and intervention launch were staggered, which allowed for limited resources to be used strategically. We have added this information to the manuscript on p.6, design section.

4. It is not uncommon that at least one statistician in the trial team is not blind to the randomisation. The justification given (the complexity of the analysis) is however unusual. I assume that the authors mean that one statistician will remain un-blind to be able to produce interim analysis reports by clusters and sequence. Please clarify.

RESPONSE: Yes, exactly. We have revised the narrative to include specific detail on why the statistician is not blind to analyses, given their role in running interim analyses by cluster and step.

5. More details are needed to describe the creation of randomisation schedules. What kind of randomisation will be employed? The number of clusters available is limited and therefore simple forms of randomization may not achieve balance between intervention and control arms at either the cluster- or participant-level.

RESPONSE: The randomization scheme was a simple cluster-level randomization, given the pragmatic nature of the design. We felt this was a reasonable choice given that the restricted age range and similar setting (university-based healthcare setting) were expected to make the clusters relatively homogenous. We have added this language to the randomization section on p.7. 

6. Unclear why such a small OR of 1.5 was used to calculate statistical power. The pilot study indicates that the magnitude of the effect is likely to be a lot larger and up to 14-fold difference. Also unclear how the individual correlation (r=0.5) and cluster-correlation (r?? <0.5) estimates are coming from.

RESPONSE: The OR=1.5 was chosen because the pilot trial was a non-randomized and non-controlled trial with a very small sample size. Our prior randomized trial of effect of text messaging on medication adherence (a related outcome) among youth living with HIV in the US found an effect size of 2.12, with a 95% CI of 1.01 – 4.45 (see PMID: 26362167).

Therefore, we chose an effect size that we felt would be as conservative as possible given the stepped wedge design. We added additional details on the cluster level correlation to the power section. Cluster correlation (ICC = 0.10) was based on prior cluster randomized trials of viral suppression interventions which generally find cluster level ICCs of 0.05 to 0.15. Individual autocorrelation were chosen based on high rates of autocorrelation among individuals in prior studies. We also note that power estimates are only marginally impacted by these parameters and, given the current estimate of power and conservative estimate for the effect size, these assumptions do not substantially impact the overall evaluation of the statistical power of the study. 

7. Statistical analysis section is quite short and has little details/clarity. Because of the use of GEE methods my understanding is that the strategy will be compared only within clusters (so that each participant act as his/her own control). Will be any comparison between sequences? If so, what is going to be the approach to analysis? Clusters can be considered independent and therefore GEE should not be needed.

RESPONSE: The GEE approach is used to control for the correlation of viral suppression across multiple observations of the same individuals over time while a fixed effect will be used to account for the effect of cluster. The main outcome is the difference in viral suppression between periods during the intervention compared to those not during the intervention, using a fixed effect for observation periods to control for secular trends in viral suppression. We have modified the analysis description to clarify these points.

8. The trial is going to be costly. I am not an expert in the field but is there a plan to evaluate whether the possible benefit in terms of increased number of youths with a suppressed viral load as a consequence of the intervention is going to be cost-effective? There is only a hint to this possible analysis in the last paragraph of the Discussion.

RESPONSE: Yes, we received supplemental funding to do a cost effectiveness analysis following completion of the trial. Because the focus of this manuscript is on the primary efficacy findings, details of the cost effectiveness evaluation are not included. 

9. One of the possible advantages of publishing a trial protocol is aiming to pre-register the study with a journal prior to data collection/collation, to avoid publication bias and limit researcher degrees of freedom. Authors should clarify whether this is an intent as it would strengthen the submission.

RESPONSE: Data collection in this study is on-going, thus this protocol manuscript will not be published prior to data collection. However, this manuscript is a more detailed version of the clinicaltrials.gov protocol registration, which was submitted in parallel with the start of the study. 

Minor points

1. Wording is fairly confusing with some of the sentences phrased in the future ‘it is anticipated that…’ ‘participants will be….’ ‘quality checks will be conducted’ ‘data will be analysed’ etc. vs. others which are placed in the past (like the trial has actually already occurred) e.g. ‘Cluster 1 was not randomised’ ‘participants were recruited’ etc. Suggest phrasing is standardised throughout.

RESPONSE: The differences in tense reflect whether or not the activity has occurred. The manuscript has been reviewed to ensure that the tenses used reflect actual initiation (or not) of the activities described. 

2. Lines 90-96. Form the way the paragraph is structured, it is not obvious at first reading that these are the results of a separate pilot study and not those produced by preliminary data of the current protocol.

RESPONSE: We have revised line 91 of the manuscript to make it clear that the UG3 phase of the study was completed prior to the initiation of the protocol described in this manuscript. 

3. Line 95. Please indicate exact p-value instead of <0.05.

RESPONSE: Exact p-values have been added as suggested. 

Reviewer #2: The authors present a description of the methods for a cluster randomized stepped wedge trial of an intervention to increase viral suppression on HIV-1 antiretroviral treatment targeted towards youth. The protocol is clear and well written and is presented in a manner that others could replicate. I have no substantive edits for the authors. Thanks for following the SPIRIT guidelines.

Minor comment:

Based on the results of the pilot, it would appear the trial is overpowered if the target is 1.5 times the odds of viral suppression comparing pre to post implementation. The authors might want to consider what is a clinically important increase in viral load suppression (although anything up from the baseline in the 30's would be better) rather than the 1.5 which would not result viral suppression levels that are beneficial at a the target population level.

RESPONSE: The OR=1.5 was chosen because the pilot iCARE UG3 trial was a non-randomized and non-controlled trial with a very small sample size. Our prior randomized trial of effect of text messaging on medication adherence (a related outcome) among youth living with HIV in the US found an effect size of 2.12, with a 95% CI of 1.01 – 4.45 (see PMID: 26362167).

It appears the population for the intervention is all youth regardless of mode of acquisition of HIV infection. Do the authors anticipate any difference in response to the intervention by youth who were perinatally infected versus infected as adolescents or young adults?

RESPONSE: In our prior trial of text messaging on medication adherence in the U.S. and in the UG3 iCARE pilot trial, we did not find significant differences among vertically versus horizontally infected youth. However, we expect to explore these potential moderation effects in this trial as well.

7. PLOS authors have the option to publish the peer review history of their article (what does this mean?). If published, this will include your full peer review and any attached files.

Do you want your identity to be public for this peer review? For information about this choice, including consent withdrawal, please see our Privacy Policy.

Reviewer #1: No

Reviewer #2: No

---

## [Decision Letter · Decision Letter 1]

31 May 2023

A randomized stepped wedge trial of an intensive combination approach to roll back the HIV epidemic in Nigerian Adolescents: iCARE Nigeria treatment support protocol

PONE-D-22-22897R1

Dear Dr. Kuhns,

We’re pleased to inform you that your manuscript has been judged scientifically suitable for publication and will be formally accepted for publication once it meets all outstanding technical requirements.

Kind regards,

Joel Msafiri Francis, MD, MS, PhD

Academic Editor

PLOS ONE

Additional Editor Comments (optional):

Reviewers' comments:

Reviewer's Responses to Questions

**Comments to the Author**

1. Does the manuscript provide a valid rationale for the proposed study, with clearly identified and justified research questions?

Reviewer #2: Yes

2. Is the protocol technically sound and planned in a manner that will lead to a meaningful outcome and allow testing the stated hypotheses?

Reviewer #2: Yes

3. Is the methodology feasible and described in sufficient detail to allow the work to be replicable?

Reviewer #2: Yes

4. Have the authors described where all data underlying the findings will be made available when the study is complete?

Reviewer #2: Yes

5. Is the manuscript presented in an intelligible fashion and written in standard English?

Reviewer #2: Yes

6. Review Comments to the Author

You may also provide optional suggestions and comments to authors that they might find helpful in planning their study.

Reviewer #2: The authors have addressed my comments. I think the effect size used in the sample size calculation is still too conservative given the use of an odds ratio as the measure of effect but the authors have adequately addressed my question.

7. PLOS authors have the option to publish the peer review history of their article (what does this mean?). If published, this will include your full peer review and any attached files.

Reviewer #2: No

---

## [Editor Report · Acceptance letter]

29 Jun 2023

PONE-D-22-22897R1 

A randomized stepped wedge trial of an intensive combination approach to roll back the HIV epidemic in Nigerian Adolescents:  iCARE Nigeria treatment support protocol 

Dear Dr. Kuhns:

I'm pleased to inform you that your manuscript has been deemed suitable for publication in PLOS ONE. Congratulations! Your manuscript is now with our production department. 

Kind regards, 

on behalf of

Dr. Joel Msafiri Francis 

Academic Editor

PLOS ONE